# EuroSpeech: A Multilingual Speech Corpus

**Samuel Pfisterer**     **Florian Grötschla**     **Luca A. Lanzendörfer**
**Florian Yan**     **Roger Wattenhofer**

ETH Zurich
`{spfisterer, fgroetschla, lanzendoerfer, floyan, wattenhofer}@ethz.ch`

## Abstract

Recent progress in speech processing has highlighted that high-quality performance across languages requires substantial training data for each individual language. While existing multilingual datasets cover many languages, they often contain insufficient data for most languages. Thus, trained models perform poorly on the majority of the supported languages. Our work addresses this challenge by introducing a scalable pipeline for constructing speech datasets from parliamentary recordings. The proposed pipeline includes robust components for media retrieval and a two-stage alignment algorithm designed to handle non-verbatim transcripts and long-form audio. Applying this pipeline to recordings from 22 European parliaments, we extract over 61k hours of aligned speech segments, achieving substantial per-language coverage with 19 languages exceeding 1k hours and 22 languages exceeding 500 hours of high-quality speech data. We obtain an average 41.8% reduction in word error rates over baselines when finetuning an existing ASR model on our dataset, demonstrating the usefulness of our approach.

## 1 Introduction

Multilingual speech datasets have been essential for the recent progress in automatic speech recognition (ASR) and text-to-speech (TTS) model performance gains. The most significant advancements in ASR and TTS rely on large-scale training data, which is available for only a handful of high-resource languages. For the vast majority of languages, the lack of transcribed speech data poses a major obstacle to building reliable models. Recent large-scale ASR work [25], suggests that at least 1k hours of transcribed speech per language is typically required for modern ASR systems to reach acceptable performance. Table 1 compares several major multilingual speech datasets, illustrating a persistent imbalance. While some datasets span over 100 languages, only a small subset provide more than 1k hours of data for more than a few languages. For example, Common Voice [2] includes over 130 languages but only 8 of them exceed 1k hours. VoxPopuli [31], a benchmark dataset from parliamentary speech, includes 16 languages but none meet this threshold. This imbalance limits the ability of multilingual models to generalize well beyond a small set of dominant languages. Parliamentary proceedings present a compelling source of multilingual speech. Many governments make recordings and transcripts of their sessions publicly available, offering long-form speech in their official language. Moreover, most national parliaments provide more than 1k hours of data. However, building usable speech datasets from these sources is complex: The data is typically fragmented across platforms and formats, transcripts are often non-verbatim, and recordings are long and unsegmented. Current pipelines for dataset construction are limited to clean transcripts, which makes it difficult to scale across many parliaments, as individual transcript cleaning would be required.

In this work, we address these limitations through two primary contributions. First, we present a scalable, open-source pipeline for constructing speech datasets from parliamentary proceedings.

39th Conference on Neural Information Processing Systems (NeurIPS 2025) Track on Datasets and Benchmarks.

Table 1: Comparison of Major Multilingual Speech Datasets. ">1k hrs" and ">500 hrs" columns indicate the number of languages exceeding these data volume thresholds. While some datasets contain more languages, EUROSPEECH provides the most languages above the respective data volume thresholds. The amount of languages over 500 and 1k hrs for the USM dataset is unknown.

| Dataset | # Languages | Total Hours | >1k hrs | >500 hrs | Availability |
|---|---|---|---|---|---|
| MSR-86k [16] | 15 | 86.3k | 14 | 15 | Public |
| Common Voice [2] | 133 | 22.1k | 8 | 15 | Public |
| MLS [23] | 8 | 50.0k | 4 | 6 | Public |
| FLEURS [4] | 102 | 1.4k | 0 | 0 | Public |
| YODAS [17] | 149 | 369.5k | 13 | 15 | Public |
| Emilia [10] | 6 | 101.0k | 5 | 6 | Public |
| VoxPopuli [31] | 16 | 1.8k | 0 | 0 | Public |
| GigaSpeech 2 [32] | 3 | 30.0k | 3 | 3 | Public |
| Whisper Data [25] | 91 | 680.0k | 16 | 21 | Private |
| BABEL [6] | ~26 | ~1.0k | 0 | 0 | Private |
| USM [34] | 73 | 90.0k | n/a | n/a | Private |
| MMS-Lab [24] | 1107 | 44.7k | 0 | 0 | Private |
| **EuroSpeech** | **22** | **61k** | **19** | **22** | **Public** |

The pipeline includes tools for downloading media and transcript files from diverse sources, as well as a robust alignment module featuring a two-stage dynamic alignment algorithm. The system supports various transcript formats (e.g., PDF, DOCX, SRT) via built-in, extensible parsers, and performs transcript normalization alongside optional LLM-based cleaning. It is specifically designed to be robust against non-verbatim transcripts and easily adaptable to new data sources with minimal engineering effort. Second, we introduce EUROSPEECH,[1] a new multilingual speech dataset built using our pipeline. EUROSPEECH comprises approximately 61k hours of aligned speech across 22 languages. Based on filtering by character error rate (CER), we obtain high-quality subsets including 50.5k hours at CER < 20%. This subset provides over 1k hours of data for 19 languages and over 500 hours for 22 languages. Together, these contributions provide a versatile pipeline for multilingual dataset creation and a valuable new resource for training and evaluating ASR and TTS models across a wider range of languages.

## 2 Related Work

### 2.1 Multilingual Speech Datasets

Initial ASR research featured single-language datasets such as Switchboard [9] and LibriSpeech [22], before progressing to multilingual efforts such as the IARPA BABEL program for low-resource languages [6]. Public multilingual corpora expanded with audiobook-derived MLS [23], the broadly crowdsourced Common Voice [2], and the benchmarking-focused FLEURS dataset [4]. Web-sourced data further increased scale in datasets such as YODAS [17], MSR-86k [16], GigaSpeech 2 [32], and Emilia [10], the last emphasizing diverse spontaneous speech.

Despite these advancements and large total reported hours, a critical imbalance persists: the large majority of languages in publicly available datasets contain below 1k hours of data per language, as detailed in Table 1. This skewness limits the development of high-performing multilingual ASR systems across many languages. The significant scale of private datasets (e.g., for Whisper [25], Google's USM [34], Meta's MMS-Lab [24]) highlights the benefits of extensive data but their inaccessibility underscores the need for large, open datasets.

Parliamentary speech, used in datasets such as VoxPopuli [31], Europarl-ASR [7], and various national corpora [14, 15, 30, 26, 11, 27, 8, 18], offers a promising domain-specific source of multilingual data. However, VoxPopuli, the largest publicly available multilingual parliamentary dataset only contains 1.8k hours across 16 languages, none of them exceeding the 1k hours threshold recommended for robust ASR training.

---

[1]EUROSPEECH is available at `https://huggingface.co/datasets/disco-eth/EuroSpeech`

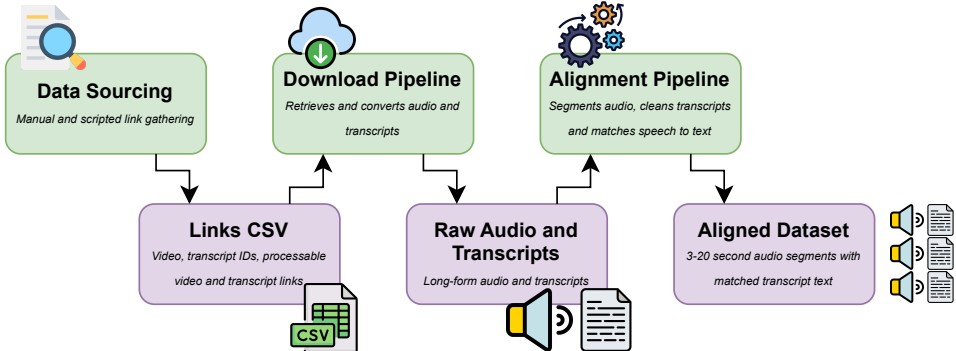

Figure 1: Overview of the EUROSPEECH data processing pipeline. The workflow begins with the initial **Data Sourcing and Metadata Collection** phase, which gathers metadata from parliamentary websites and APIs. This structured information (as **Links CSV**) feeds into the **Download Pipeline** to retrieve raw audio and transcripts. The **Raw Audio and Transcripts** are then processed by the **Alignment Pipeline**, which segments the audio and matches it to the corresponding text. The final output is the **Aligned Dataset**, consisting of short audio segments paired with their transcriptions, ready for model training.

EUROSPEECH differs from VoxPopuli in two key aspects. First, the scale and language coverage differ substantially: while VoxPopuli contains 1.8k hours with no languages exceeding 1k hours of transcribed data, EUROSPEECH provides 50.5k hours (CER < 20%) with 19 languages exceeding the 1k hour threshold. Second, the data sources differ fundamentally. VoxPopuli collected data from the European Union Parliament in Brussels, where representatives of all EU nations give speeches in their various languages, with recordings and transcripts sourced from a single website. In contrast, EUROSPEECH gathers parliamentary speeches from each country's individual national parliament, requiring custom scripts to collect data from 22 separate parliamentary websites.

Our work directly addresses these gaps by significantly increasing both the scale and language coverage of publicly available parliamentary speech data, as demonstrated by our EUROSPEECH dataset (Table 1).

### 2.2 Speech-Text Alignment Techniques

The construction of speech datasets requires precise alignment between raw audio and textual transcripts, which becomes particularly complex when processing noisy or non-verbatim sources such as parliamentary recordings. Alignment pipelines typically follow several stages: initial segmentation (via voice activity detection, speaker diarization, or fixed-duration chunking), matching audio segments to transcript sections (often through approximate ASR-based text matching), forced alignment (FA) for precise time-stamping, and finally, refined segmentation and quality filtering [23, 3, 31, 33, 24, 22, 7]. Some recent datasets (e.g., LibriHeavy [13]) used off-the-shelf ASR models such as Whisper for ASR-based audio-text matching. Notably, bootstrapping methods that use preliminary alignments to train better acoustic models for subsequent dataset re-alignment have proven effective [3, 24].

Our alignment pipeline incorporates a two-stage dynamic alignment approach that effectively handles noisy, non-verbatim transcripts typical of parliamentary speech, requiring minimal manual intervention. The approach is inspired by the dynamic CER-based matching employed by the VAC pipeline [33], yet extends it to handle the non-verbatim and noisy parliamentary transcripts robustly.

## 3 Data Collection Process

Building high-quality multilingual speech datasets from real-world parliamentary data introduces unique challenges: audio and transcripts are often long, noisy, and poorly aligned; metadata is

inconsistent or absent; and content is served through a wide range of web technologies. To address these issues, we design a scalable, modular system that transforms raw parliamentary recordings into clean, aligned speech-text segments suitable for model training. Our design prioritizes extensibility, fault isolation, and low operational overhead, enabling non-expert teams to replicate this process for new languages and data sources.

## 3.1 Pipeline Overview

Our proposed pipeline, shown in Figure 1, comprises an initial data sourcing and metadata collection phase, followed by two core automated pipelines: The **Download Pipeline** retrieves, standardizes, and converts raw audio and transcripts using a format-agnostic architecture. The **Alignment Pipeline** segments, transcribes, and aligns multi-hour recordings with non-verbatim transcripts using a noise-tolerant, dynamic matching algorithm.

These stages automatically process parliamentary data formats with minimal manual intervention. The two-stage alignment algorithm (details in Section 3.4) robustly handles heterogeneous data sources and non-verbatim transcripts, requiring only initial data source links as input. Once these links are collected, the pipeline performs all downloading, processing, and alignment steps automatically, eliminating the need for format-specific customization or manual transcript pre-processing.

## 3.2 Data Sourcing and Metadata Collection

Parliamentary data is published in a wide range of formats with little standardization across countries. Some parliaments provide direct access to downloadable MP4 files and clean transcripts in HTML; others publish only streaming video behind dynamic players or offer transcripts as scanned PDFs or DOCX documents. The goal of this initial data sourcing stage is to overcome the challenge of these diverse and unstandardized parliamentary sources. This is achieved by extracting and organizing key metadata (e.g., media and transcript URLs) into a standardized CSV file. This CSV then provides a consistent input format, enabling the subsequent automated download and alignment pipelines to operate uniformly, irrespective of the origin of the data.

The process begins with manual inspection of each parliament website, these can range from dedicated media portals to various scattered web pages. We identify data formats, access methods, and the structure of session-related information. This investigation, which can be challenging due to inconsistencies in how data is published and linked, informs the development of custom data collection scripts. These scripts aim to extract metadata for each session, which typically includes an audio or video URL, links to one or more potential transcripts, and a unique session identifier. We store this information in a standardized CSV format. The resulting CSV serves as the interface between this initial collection phase and the subsequent phases of the pipeline. By encapsulating all source-specific access logic and discovered links into this metadata file, the download and alignment pipelines can operate uniformly, allowing for a simple and efficient way of scaling to new parliaments.

## 3.3 Download Pipeline

Given the structured CSVs produced by the initial data sourcing and metadata collection stage, the download pipeline automates the retrieval of referenced audio, video, and transcript files. This stage is designed for robustness and extensibility across various types of content. The pipeline ingests diverse source types through a dispatch architecture, mapping URLs to specialized handlers. Built-in handlers cover common sources (e.g., direct links, YouTube, dynamic pages). Custom extractors handle parliamentary websites that require custom link extraction or transcript processing (e.g., special video players, multi-page transcript collection) without the need to change the core logic. Additionally, we implemented checkpointing, error recovery, and parallelization, as well as session-level download status, tracked in a central PostgreSQL database. These features enable distributed, non-redundant execution and automatic retries for failed downloads.

## 3.4 Alignment Pipeline

The alignment pipeline transforms raw audio and transcripts into short paired segments suitable for ASR and TTS model training. This stage addresses the challenges of long, noisy audio and diverse, non-verbatim transcripts as well as ambiguous mappings to create high-quality training data. Given

a long input recording (often between 1–10 hours) and a set of candidate transcripts (e.g., in PDF, HTML, or DOCX format), the pipeline first segments the audio into 3–20 second utterances using voice activity detection [28]. Each segment is then transcribed using an ASR model. We use Whisper Turbo [25] as the default ASR model, in the case of Maltese we use a fine-tuned Whisper Model [12]. These generated ASR text snippets are used to align the audio segments to the human transcript. Additionally, our proposed pipeline supports any ASR model as well as speaker diarization, which is useful to ensure single speaker audio segments. Raw transcripts are automatically preprocessed into cleaned and standardized text. The pipeline includes built-in parsers for common formats (e.g., PDF, DOCX, HTML, TXT, SRT), with optional LLM-based cleaning to remove non-speech elements from documents such as PDFs (cf. Appendix D). This stage of the pipeline is easily extensible for new formats or custom logic.

We propose a **two-stage dynamic algorithm** to align ASR generated transcripts with parliamentary transcripts (detailed algorithm pseudo-code in Appendix B). The parliamentary transcripts often contain speech content mixed with non-verbatim text, unuttered text, or speaker annotations. This algorithm enables data collection from diverse sources by eliminating the need for manual transcript pre-processing. Unlike VAC [33], which uses a single dynamic window for alignment, our approach uses a two-stage coarse-to-fine strategy to identify matching transcript text for each audio segment:

1. **Coarse Search:** Uses a sliding window of size $n$, where $n$ is the length of the current ASR generated transcript. The starting position for the sliding window is set to the last matched position of the previous segment. This coarse search identifies candidate text spans by computing the CER between each transcript window and the ASR generated transcript, skipping transcript sections with high error rates. We pass the first candidate that has a CER below 30% to the refined search stage. If no candidates below 30% CER are found, we pass the k (default k = 3) candidates with the lowest CER to the refined stage.

2. **Refined Search**: The coarse candidate windows identified in the first stage are adjusted to find the lowest possible CER within the respective window. For each candidate window, we test different start positions and window sizes within a local margin. Specifically, the start position is varied within $\pm 15$ words around the coarse window's starting position, and for each start position, we test window sizes ranging from $(L - 15)$ to $(L + 15)$ words, where $L$ represents the ASR segment length in words. We select the start position and window size that minimizes CER between the transcript span and ASR generated transcript.

A fallback mechanism restarts this two-stage search for the current segment. The key difference is that the starting position for the coarse search is set to the beginning of the entire transcript rather than the last matched position from the previous segment. The fallback mechanism addresses cases where previous misalignments positioned the starting position beyond the current segment's true location in the transcript. Restarting from the beginning allows the algorithm to find the correct match instead of searching past it.

Our alignment algorithm requires pairing each audio file with a single transcript file. However, parliaments may provide multiple transcript files for a given day, and it is often not clear which of these transcripts belongs to which audio file. For example, when transcripts and audio files can only be matched by their publication date and multiple files share the same date, we cannot clearly determine which transcript belongs to which audio. Additionally, individual transcripts are often available in multiple formats (e.g., PDF, HTML, DOCX). To automate finding the correct pairings, we use a two-stage selection process: we first align the audio to all candidate transcripts in all available formats, then select the best format for each transcript based on median CER, and finally choose the transcript to use based on configurable criteria (e.g., lowest CER or all transcripts below a CER threshold).

The output of this alignment stage includes a JSON summary for each audio file and detailed alignment files for each selected transcript. Each file contains timestamps, matched text, and quality metrics. These files form the basis for downstream filtering and dataset generation.

## 3.5  Filtering and Output Format

Once segment-level alignments have been established, the final step is to filter and package the aligned data for training. Although the alignment algorithm produces timestamps and CER estimates for each

Table 2: Overview of EUROSPEECH: Aligned audio hours per language at our three quality character error rate (CER) Thresholds. Languages are sorted by hours at CER < 20% which represents the main subset of EUROSPEECH.

| Language | Code | Total Aligned (h) | CER < 30% (h) | CER < 20% (h) | CER < 10% (h) |
|---|---|---|---|---|---|
| Croatia | hr | 7484.9 | 5899.7 | 5615.8 | 4592.0 |
| Denmark | da | 7014.2 | 6435.0 | 5559.8 | 3443.7 |
| Norway | no | 5326.2 | 4578.8 | 3866.7 | 2252.2 |
| Portugal | pt | 5096.3 | 4036.7 | 3293.5 | 2105.9 |
| Italy | it | 4812.8 | 3539.6 | 2813.7 | 1767.3 |
| Lithuania | lt | 5537.9 | 3971.0 | 2681.2 | 956.6 |
| United Kingdom | en | 5212.2 | 3790.7 | 2609.3 | 1175.0 |
| Slovakia | sk | 2863.4 | 2722.4 | 2553.6 | 2070.8 |
| Greece | el | 3096.7 | 2717.6 | 2395.4 | 1620.9 |
| Sweden | sv | 3819.4 | 2862.6 | 2312.8 | 1360.1 |
| France | fr | 5476.8 | 2972.1 | 2249.8 | 1347.6 |
| Bulgaria | bg | 3419.6 | 2570.4 | 2200.1 | 1472.8 |
| Germany | de | 2472.2 | 2354.2 | 2184.4 | 1698.4 |
| Serbia | sr | 2263.1 | 1985.1 | 1855.7 | 1374.1 |
| Finland | fi | 2130.6 | 1991.4 | 1848.2 | 1442.2 |
| Latvia | lv | 2047.4 | 1627.9 | 1218.8 | 499.9 |
| Ukraine | uk | 1287.8 | 1238.3 | 1191.1 | 1029.8 |
| Slovenia | sl | 1338.2 | 1241.7 | 1156.4 | 900.5 |
| Estonia | et | 1701.1 | 1430.9 | 1014.9 | 382.5 |
| Bosnia & Herz. | bs | 860.2 | 781.9 | 691.3 | 447.8 |
| Iceland | is | 1586.1 | 974.1 | 647.4 | 171.4 |
| Malta | mt | 3281.6 | 1284.3 | 613.0 | 143.9 |
| **Total** | | **78128.6** | **61006.4** | **50572.9** | **32255.5** |

utterance, not all matched pairs are equally reliable, especially given transcript noise, overlapping speech, or ASR errors.

We adopt CER-based filtering as the primary mechanism for quality control. A threshold (e.g., CER < 20%) is applied for each segment, allowing users to select a desired trade-off between dataset size and quality. Additionally, we track total aligned duration at multiple quality tiers (e.g., CER < 10%, < 20%, < 30%), enabling granular evaluation and filtering without rerunning the alignment pipeline.

For each audio file we output a summary JSON file containing overall alignment statistics and references to all candidate transcripts. For each accepted audio-transcript pair, we generate a separate alignment JSON file listing all aligned segments with their start and end times, ASR text, matched human transcript text, and CER.

## 4 The EUROSPEECH Dataset

Running our proposed data processing pipeline described in Section 3, we constructed EUROSPEECH, a new multilingual speech corpus derived from parliamentary proceedings from 22 European nations. To create EUROSPEECH we processed as many publicly available parliamentary recordings as possible. After the initial alignment and segmentation process, which removed silences and unaligned portions, we obtained approximately 78k hours of aligned speech-text data. Finally, we then created quality-filtered subsets based on Character Error Rate (CER):

- CER < 30%: approximately 61k hours (78.2% of aligned data)

- CER < 20%: approximately 51k hours (65.4% of aligned data)

- CER < 10%: approximately 32k hours (41.0% of aligned data)

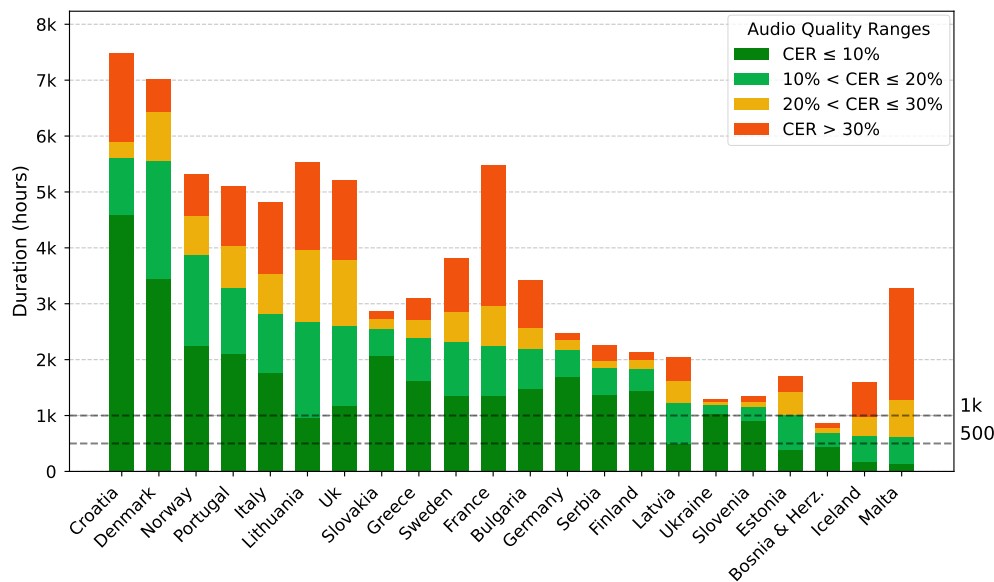

Figure 3: Audio duration for each language in EUROSPEECH after the alignment pipeline and at different Character Error Rate (CER) filtering stages (CER < 30%, < 20%, and < 10%). Languages are ordered by their data volume at CER < 20%. A dashed horizontal lines indicates the 1,000-hour and 500-hour thresholds. EUROSPEECH showcases large amounts of low-CER data across its languages.

The CER < 20% subset serves as our primary dataset for comparisons, we chose the 20% threshold based on VoxPopuli [31]. Within this subset, EUROSPEECH provides >1k hours of data for 19 languages and >500 hours for 22 languages. Table 2 presents a detailed breakdown of the dataset composition for each language.

The audio segments in EUROSPEECH typically range from 3 to 20 seconds, durations suitable for training ASR and TTS models. The audio in the published dataset is sampled at 16 kHz, which is the standard sampling rate for training ASR models. We plan to upload a 24 kHz version of the dataset as this is most common for TTS models. The data reflects the formal speaking style characteristic of parliamentary debates. The European coverage of the data is shown in Figure 2.

Unlike many existing multilingual datasets that are heavily skewed toward a few high-resource languages, EUROSPEECH maintains a more equitable distribution across languages, a key differentiator of our corpus. Figure 3 further details the data quantities per language and the impact of CER filtering stages per language.

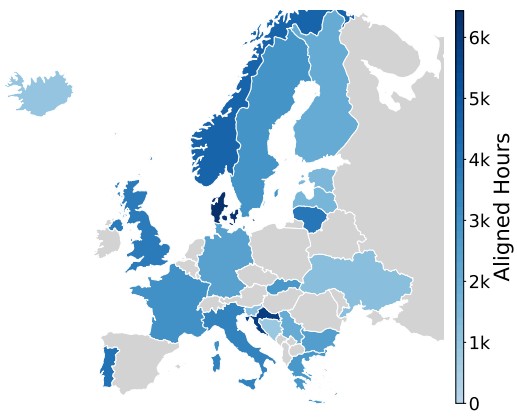

Figure 2: The broad European geographic coverage of EUROSPEECH. Countries are colored by the total hours of aligned speech data (CER < 30%) available in the dataset. A perceptually uniform color scale is used.

To facilitate standardized benchmarking, we provide predefined train, development, and test splits for each language. To ensure data integrity and prevent leakage between sets, these splits are constructed by assigning entire parliamentary sessions (i.e., all segments derived from a single original long audio recording) exclusively to one of the train, development, or test sets. The proportions for these splits follow common practices and are detailed in the dataset repository.

Table 3: Word Error Rates (%) of Whisper v3 Turbo on the FLEURS test set before and after finetuning on EUROSPEECH. Finetuned models consistently reduce WER across all evaluated languages, demonstrating the practical value of the dataset for improving multilingual ASR performance.

| Language | Baseline | Finetuned | Rel. Improvement |
|----------|----------|-----------|------------------|
| Maltese | 72.2 | 25.9 | 64.1% |
| Icelandic | 20.0 | 15.0 | 25.0% |
| Lithuanian | 25.0 | 15.9 | 36.4% |
| Latvian | 19.3 | 11.1 | 42.5% |
| Slovenian | 20.5 | 13.0 | 36.7% |
| Estonian | 18.4 | 9.9 | 46.1% |
| Average | 29.2 | 15.1 | 41.8% |

## 5 Finetuning ASR Models with EUROSPEECH

This section evaluates the effectiveness of our EUROSPEECH dataset for improving ASR performance, particularly on low-resource European languages. We demonstrate that finetuning pretrained multilingual ASR model Whisper v3 Turbo on the EUROSPEECH dataset yields considerable improvements in transcription performance.

### 5.1 Experimental Setup

We evaluate the impact of our dataset by finetuning the pretrained Whisper v3 Turbo model [25] on six European languages from our collection: Maltese, Icelandic, Lithuanian, Latvian, Slovenian, and Estonian. These languages were selected to represent different language families and focus on low-resource availability levels in the existing literature.

**Baseline Model**: We use Whisper v3 Turbo as our baseline for finetuning. This model was pretrained on 680k hours of labeled audio in 98 languages but has limited exposure to many European languages in our dataset. As of the time of writing, performance on the FLEURS [4] test set was not publicly reported for certain languages targeted in our finetuning. Consequently, we conducted independent baseline evaluations of the Whisper v3 Turbo model on FLEURS.

**Finetuning**: For each of the six languages we evaluated, approximately 200 hours of training data with the lowest CER were selected for training. More details can be found in Appendix A.

**Evaluation**: We evaluate the models on the FLEURS test set. For evaluation metrics, we report the Word Error Rate (WER). To ensure fair comparison across languages with different writing systems and word formation patterns, we apply the NFKC normalization process as used in the training of all Whisper models before computing error rates.

### 5.2 Results

Table 3 presents the performance of the finetuned Whisper v3 Turbo compared to the baseline on the FLEURS test set. We observe that, finetuning on EUROSPEECH substantially improves transcription performance across all tested languages. On average, we observe a relative WER reduction of 41.8% on the out-of-domain FLEURS test set. The results demonstrate that our dataset enables significant WER improvements, achieving competitive performance for open-source ASR models, while using a limited subset of EUROSPEECH's training data.

## 6 Limitations

While EUROSPEECH represents a balanced dataset containing various low-resource languages, certain limitations remain. The dataset is derived entirely from parliamentary recordings, a domain characterized by formal, planned, and often repetitive speech. This linguistic register, while useful for certain modeling tasks, may not adequately represent the diversity of natural spoken language encountered in more spontaneous or informal settings. As such, models trained solely on EuroSpeech may exhibit reduced performance when deployed in conversational or non-scripted speech scenarios.

The dataset's linguistic and geographic scope, although broad within the European context, remains limited in global coverage. Many underrepresented languages, particularly those outside of Europe, are not included. Even within the covered languages, variation in dialect, sociolect, and regional accents is likely constrained by the nature of parliamentary speech, which tends to reflect standard or official varieties. This may affect the robustness and fairness of models trained on the dataset, particularly in settings requiring sensitivity to linguistic diversity.

From a technical perspective, the alignment process is dependent on the quality of existing ASR models, which are used to generate intermediate transcriptions that are needed to align the human transcript with the correct audio segment. While the proposed alignment algorithm is designed to be robust to non-verbatim transcripts and noisy inputs, its performance is ultimately constrained by the capabilities of the underlying ASR models, which can vary significantly across languages and acoustic conditions. In low-resource languages or in instances of degraded audio quality, the accuracy of alignments may be reduced, potentially impacting the quality of the resulting training data.

# 7 Conclusions

In this work, we presented a source-agnostic, open-source pipeline for speech-text alignment that can process any audio with potentially matching transcripts. Its robust two-stage alignment algorithm and modular architecture enables non-experts to create high-quality speech datasets for diverse applications and languages. Using our proposed pipeline, we created EUROSPEECH, a multilingual speech corpus containing over 50.5k hours of aligned parliamentary speech with CER < 20% across 22 European languages. Unlike existing public datasets, EUROSPEECH provides substantial data for all included languages, with 19 languages exceeding 1k hours and 22 exceeding 500 hours. The balanced distribution across languages addresses the severe imbalance in current multilingual speech resources. To demonstrate the usefulness of our dataset, we finetune an ASR model on EUROSPEECH for six low-resource languages, showing an average 41.8% reduction in word error rates over baselines. The pipeline codebase as well as the dataset are made publicly available.

We believe that both the EUROSPEECH dataset and our proposed pipeline can be useful starting points for further work on multilingual and low-resource speech processing. In the future, the pipeline could be extended to other domains such as conversational speech, or adapted to include more languages and metadata such as speaker or session information. Making these tools available can help lower the barrier to building high-quality datasets, especially for underrepresented languages.

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

# A Experiment Setup

**Data Collection Infrastructure.** Download jobs were allocated 2 CPUs and 8GB RAM each. The total computational cost for data sourcing comprised approximately 4,200 hours across video downloads (3,930 hours) and transcript retrieval (280 hours). These estimates are based on job logging in our database and provide approximate resource requirements for replication.

**Alignment Pipeline Compute.** Audio-text alignment was performed using heterogeneous GPU resources including GeForce RTX 2080 Ti (11GB), Tesla V100 (32GB), Titan XP (12GB), Titan RTX (24GB), and RTX 3090 (24GB). The majority of computation utilized RTX 2080 Ti and Tesla V100 cards. Total alignment processing required approximately 5,548 GPU-hours across all jobs and languages.

**ASR Model Fine-tuning.** We fine-tuned Whisper v3 Turbo[2] using the following hyperparameters: batch size 64, gradient accumulation steps 2, learning rate 1e-5, warmup ratio 0.06, and linear learning rate scheduling. Training details for each language are provided in Table 4. All trainings were performed on NVIDIA RTX A6000 GPU cards.

Language selection was motivated by two factors: (1) these six languages exhibited the highest baseline WER with Whisper v3 Turbo, allowing demonstration of meaningful improvements with limited computational resources, and (2) poor baseline ASR performance creates additional challenges for our alignment pipeline, as ASR transcriptions for these languages contain more errors, providing a rigorous test of our pipeline's ability to match noisy ASR outputs to the correct segments in human transcripts.

Table 4: Fine-tuning configuration per language

| Language | Training Data (h) | Epochs | Training Time (h) |
|---|---|---|---|
| Maltese | 143 | 0.2 | 1.3 |
| Icelandic | 213 | 1.6 | 13.2 |
| Lithuanian | 365 | 2.5 | 43.2 |
| Latvian | 203 | 2.4 | 21.0 |
| Slovenian | 289 | 3.0 | 40.5 |
| Estonian | 262 | 2.8 | 28.6 |

---

[2] https://huggingface.co/openai/whisper-large-v3-turbo

# B  Data Collection Process

---

**Algorithm 1** Two-Stage Dynamic Alignment Algorithm

---

**Require:** ASR segments $S_{asr}$, Full Transcript Text $T$
**Require:** CER threshold $\theta$
**Ensure:** List of Aligned Segments $S_{aligned}$

1: $S_{aligned} \leftarrow \emptyset$
2: $last\_end\_idx \leftarrow 0$ $\qquad\qquad\qquad\qquad\qquad$ ▷ End index of last matched transcript segment
3: **for all** segment $s_{asr} \in S_{asr}$ **do**
4: $\qquad\qquad\qquad\qquad\qquad\qquad$ ▷ Stage 1: Coarse Search (sequential from $last\_end\_idx$)
5: $\quad$ $candidates \leftarrow$ CoarseSearch($s_{asr}, T$, start_idx=$last\_end\_idx$)
6: $\qquad\qquad\qquad\qquad\qquad\qquad\qquad$ ▷ Stage 2: Refining within candidate regions
7: $\quad$ $match \leftarrow$ RefinedSearch($candidates, s_{asr}, T$)
8: $\quad$ **if** $match.cer > \theta$ **then** $\qquad\qquad\qquad\qquad$ ▷ Fallback 1: Global Coarse Search
9: $\qquad$ $candidates_{global} \leftarrow$ CoarseSearch($s_{asr}, T$, start_idx=0)
10: $\qquad$ $match_{global} \leftarrow$ RefinedSearch($candidates_{global}, s_{asr}, T$)
11: $\qquad$ **if** $match_{global}.cer > \theta$ **then** $\qquad\qquad$ ▷ Global search also insufficient
12: $\qquad\quad$ $match \leftarrow$ DefaultMatch($s_{asr}, T, last\_end\_idx$) $\qquad$ ▷ Fallback 2
13: $\qquad$ **else**
14: $\qquad\quad$ $match \leftarrow match_{global}$ $\qquad\qquad\qquad\qquad$ ▷ Use global match
15: $\qquad$ **end if**
16: $\quad$ **end if**
17: $\quad$ Append $match$ to $S_{aligned}$
18: $\quad$ $last\_end\_idx \leftarrow match.end\_idx$ $\qquad\qquad\qquad$ ▷ Update for next sequential search
19: **end for**
20: **return** $S_{aligned}$

---

The two-stage dynamic alignment algorithm matches ASR-transcribed audio segments to corresponding segments in human transcripts. Processing segments sequentially, it maintains $last\_end\_idx$ to track transcript position and leverage temporal ordering. For each segment, coarse search employs a sliding window from the last matched position to identify candidate spans with minimal character error rate. Refined search then exhaustively searches over start position offsets and window lengths within a local margin around the candidate region to minimize character error rate. If the resulting alignment exceeds threshold $\theta$ a global search across the entire transcript is attempted. When quality thresholds cannot be met, default matching performs refined search around $last\_end\_idx$ and stores the best available match regardless of CER, ensuring complete dataset coverage.

# C  Data Sources

We sourced the parliamentary data primarily from the respective parliament websites of each country, with some additional content obtained from YouTube channels operated by the parliaments. For each country, we maintain a CSV file that lists all source links for video/audio files and transcript documents.

The video_id and transcript_id values present in the final EUROSPEECH dataset can be used to trace back to the specific source URLs for each audio segment and its corresponding text.

All CSV files containing the source metadata are publicly available on Hugging Face.[3] These files provide complete transparency regarding the origins of our dataset and enable others to replicate or extend our data collection methodology.

For copyright and licensing information regarding the parliamentary data from each country, we refer to Table 5 below, which details the relevant legal frameworks and licensing terms for each parliamentary source.

---

[3]https://huggingface.co/datasets/SamuelPfisterer1/EuroSpeech-Data-Sources

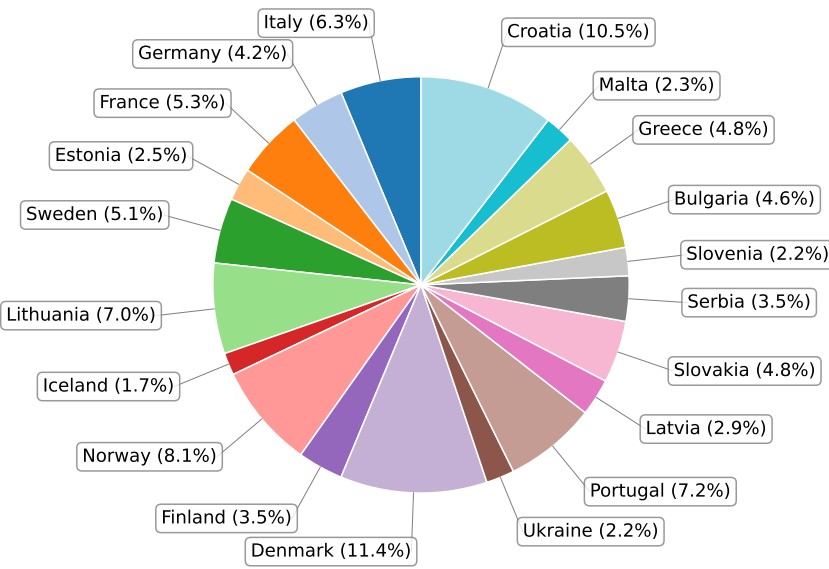

Figure 4: Language distribution in the EUROSPEECH CER < 30% subset, highlighting a key strength of our dataset: the balanced distribution across multiple languages rather than concentration in just a few dominant ones.

## D   LLM-Based Transcript Cleaning

Our pipeline provides an optional LLM-based cleaning feature specifically for PDF transcripts. When processing PDF transcripts, the pipeline first extracts text from each page. When LLM-based cleaning is used the extracted text is passed through a large language model with specific instructions to retain only spoken dialogue. We use Gemini Flash 2.0 as the default model for this cleaning step as we found it achieved a good performance-to-cost ratio.

The default system prompt used for LLM-based cleaning is:

> You are a multilingual assistant specialized in processing parliamentary transcripts. Your task is to clean the provided transcript page by removing all unnecessary metadata, annotations etc. while preserving only the literal spoken dialogue. Please follow these instructions: Remove the speaker labels that appear as headers before each speaker's dialogue. Remove all annotations, procedural notes, timestamps, and non-verbal cues. Ensure that only and all the spoken dialogue is in your response. Respond in the same language as the input and do not alter the spoken text.

The system prompt can be customized through the pipeline configuration. Tests based on one German parliamentary session showed median CER improvements from 12.3% to 9.7% for the final aligned segments when using LLM-based cleaning compared to standard PDF text extraction.

## E   Broader Impacts

This work aims to address the substantial imbalance in multilingual speech resources by introducing a large-scale, publicly available dataset with strong per-language coverage across 22 European languages. The EUROSPEECH corpus enables the development and evaluation of speech models for languages that have previously lacked sufficient training data, particularly in the context of automatic speech recognition (ASR) and text-to-speech (TTS) systems. By improving model performance

Table 5: Copyright of parliament data

| Country | Source |
|---|---|
| Croatia | Legal Notice |
| Denmark | Legal Notice |
| Norway | NLOD License |
| Portugal | Portuguese Copyright Code Article 75 |
| Italy | Italian Parliament Website references CC By 4.0 License |
| Lithuania | Republic of Lithuania Law on Copyright and Related Rights Article 22 |
| United Kingdom | Terms and Conditions for audio, Open Government Licence for transcripts |
| Slovakia | Slovak Copyright Act Chapter One Section 5e) |
| Greece | Greek Copyright Law Article 2(5) and Article 25(1)(b) |
| Sweden | Law (2022:818) |
| France | License Ouverte |
| Bulgaria | Copyright Policy references CC BY 2.5 BG |
| Germany | Terms of Use |
| Serbia | Serbian Law on Copyright and Related Rights. Article 6(2) |
| Finland | Copyright Act Article 9, 22, and 25 |
| Latvia | Latvian Copyright Law Section 21 |
| Ukraine | Law of Ukraine on Copyright and Related Rights Article 8(1)(3) |
| Slovenia | Copyright and Related Rights Act Article 46-51 |
| Estonia | Copyright Act, Estonian Youtube references CC BY SA |
| Bosnia & Herz. | Copyright Law Article 44 and 47 |
| Iceland | Copyright Act Article 22 |
| Malta | Re-Use of Public Sector Information Act Chapter 546 |

for under-resourced languages, the dataset has the potential to broaden access to speech technology and reduce the reliance on high-resource language data in multilingual systems. The dataset's origin in parliamentary recordings makes it well-suited for applications related to public sector accessibility, such as transcription and translation of government proceedings. However, this domain-specificity also imposes limitations: the speech style is formal, planned, and typically reflects standard language varieties. As a result, models trained exclusively on EUROSPEECH may generalize poorly to conversational or informal speech, and may underperform for dialectal, regional, or sociolectal variation not represented in parliamentary discourse.

The dataset is constructed from publicly available government media and does not include private or crowd-sourced content. Nevertheless, identifiable individuals may be mentioned in the transcripts, and downstream uses involving speaker identification or synthesis warrant careful consideration. While the primary goal is to support inclusive and transparent research, we acknowledge that speech models trained on EUROSPEECH could be used for purposes such as synthetic speech generation, which carries misuse potential in contexts such as impersonation or disinformation. The domain constraints of the data mitigate some of this risk, but further safeguards may be necessary depending on downstream applications.

Overall, this work contributes infrastructure that can lower the barrier to entry for multilingual speech research, particularly for low-resource languages. At the same time, it highlights the need for complementary datasets that capture greater linguistic diversity and less formal speech styles to support broader and more equitable generalization.

# F   Comparison with Existing Language-Specific Datasets

Table 6 presents a comprehensive comparison between EUROSPEECH and the largest publicly available speech datasets for each language in our corpus. This comparison demonstrates the substantial increase in available training data that EUROSPEECH provides for many European languages.

We created new state-of-the-art duration datasets for 12 languages, crossing the 1k hour threshold for 8 languages where previous datasets were below this threshold. For 5 languages, our durations are 10

Table 6: Comparison of EUROSPEECH with state-of-the-art speech datasets per language. Hours shown for EUROSPEECH correspond to the CER < 20% filtered subset. Bold values indicate cases where EUROSPEECH provides more data than existing datasets.

| Country/Language | SOTA Dataset Name | SOTA Dataset Hours | EUROSPEECH Hours |
|---|---|---|---|
| Croatia | ParlaSpeech-HR [18] | 3061 | **5615.8** |
| Denmark | FT-Speech [14] | 1800 | **5559.8** |
| Norway | Stortinget Corpus [21] | 5190 | 3866.7 |
| Portugal | MOSEL [5] | 5492 | 3293.5 |
| Italy | MOSEL [5] | 3756 | 2813.7 |
| Lithuania | Common Voice [2] | 25 | **2681.2** |
| United Kingdom | MOSEL [5] | 437238 | 2609.3 |
| Slovakia | MOSEL [5] | 61 | **2553.6** |
| Greece | YODAS [17] | 126.75 | **2395.4** |
| Sweden | RixVox-v2 [26] | 22900 | 2312.8 |
| France | MOSEL [5] | 26984 | 2249.8 |
| Bulgaria | BG-PARLAMA [8] | 249 | **2200.1** |
| Germany | MOSEL [5] | 9236 | 2184.4 |
| Serbia | ParlaSpeech-RS [18] | 896.22 | **1855.7** |
| Finland | Finnish Parliament ASR [30] | 3087 | 1848.2 |
| Latvia | Common Voice [2] | 263 | **1218.8** |
| Ukraine | YODAS [17] | 396.598 | **1191.1** |
| Slovenia | ASR database ARTUR 1.0 [29] | 884 | **1156.4** |
| Estonia | TalTech Speech Dataset [1] | 1334 | 1014.9 |
| Bosnia & Herz. | YODAS [17] | 9.37 | **691.3** |
| Iceland | Samrómur Milljón [20] | 967 | 647.4 |
| Malta | MASRI [19] | 44 | **613.0** |

to 100 times greater than those of prior state-of-the-art datasets. It is also important to highlight that some of the previous state-of-the-art datasets are not as easily usable (i.e., they do not have a unified representation and cannot be used with a few lines of code from HuggingFace). Furthermore, the quality on some of these datasets is difficult to verify as they do not explain how they filtered the dataset. The EUROSPEECH durations in this table refer to the 20% CER subset which is equivalent to the threshold used for VoxPopuli [31].

