# OpenReview forum: "EuroSpeech: A Multilingual Speech Corpus"
_NeurIPS.cc/2025/Datasets_and_Benchmarks_Track — NeurIPS 2025 Datasets and Benchmarks Track spotlight_

### Official Review · Reviewer_YWXy · 2025-06-24

**Rating:** 5
**Confidence:** 4

**Summary:**

The authors propose a pipeline for constructing a large ASR dataset from European parliament recordings. The pipeline includes components for media retrieval and two-stage alignment algorithm to handle non-verbatim transcripts and long-form audio. Overall, the dataset contains 61k hours of speech which spans across 22 languages. The pipeline consists of two stages: (1) Download pipeline, which retrieves, standardizes, and converts raw audio and transcripts using a format agnostic model. (2) Alignment pipeline, which segments, transcribes and aligns multi-hour recordings with non-verbatim transcripts using a dynamic programming based matching algorithm. The authors demonstrate that fine-tuning on the proposed dataset gives average WER reduction of 41.8% on FLEURS test set.

**Dataset Code Accessibility:**

Partly

**Ethical Considerations:**

No, there are no or only very minor ethics concerns

**Limitations Weaknesses:**

1. The dataset is not first of its kind. For example, Voxpopuli is also created from European parliamentary recordings. It would have been great if authors could have dedicated a section on describing how the proposed dataset differs from it.
2. The steps proposed for mining the dataset are quite standard so novelty wise the work is not ground breaking.
3. As highlighted by the authors themselves, the dataset covers a particular domain of parliamentary recordings so might not significantly help ASR performance in other domains such as informal speech etc.

**Strengths Contributions:**

1. The proposed dataset is large (61k hours) and can help to improve existing ASR models on European languages (22 languages).
2. Since the proposed pipeline is generic it can be continously used to expand the size of the dataset in future.

---

> ### Author Rebuttal · Authors · 2025-07-30
>
> We thank the reviewer for their thorough analysis and thoughtful feedback. Below, we address the points raised.
>
> > The dataset is not first of its kind. For example, Voxpopuli is also created from European parliamentary recordings. It would have been great if authors could have dedicated a section on describing how the proposed dataset differs from it.
>
> Our proposed dataset differs in two key aspects from the Voxpopuli dataset:
>
> 1. The hours of labelled data per language differ, as none of the languages in the voxpopuli dataset exceeds 1k hours of transcribed data, whereas our dataset has 19 languages exceeding 1k hours of  transcribed data, and the total hours of transcribed speech differ (1.8k hours for Voxpopuli vs 50k hours for Eurospeech). Additionally, note that all hours refer to speech filtered at 20% CER thresholds, both for Voxpopuli and EuroSpeech.
>
> 2. The data sources differ. The parliament of the European Union in Brussels consists of representatives of all EU nations, where these representatives give speeches and debate in their various languages. Voxpopuli gathered their data from this EU parliament, where the recordings and transcripts stem from a single website. In contrast, our approach gathers parliamentary speeches from each *countries’ individual parliament*. To this end, we wrote scripts to gather data from 22 completely separate parliamentary websites of the respective countries. This data source difference has major implications on the overall effort and the need for standardized pipelines, which we built for this work.
>
>
> > As highlighted by the authors themselves, the dataset covers a particular domain of parliamentary recordings so might not significantly help ASR performance in other domains such as informal speech etc.
>
> This is a valid concern that we covered in the section on limitations. We used the Fleurs dataset to evaluate our fine-tuned models. While not perfect, this is the default benchmarking dataset for ASR, which also does not contain parliamentary speech. These results show that fine-tuning on EuroSpeech lets the models generalize to some degree across different speech domains.

---

### Official Review · Reviewer_otqo · 2025-06-29

**Rating:** 5
**Confidence:** 4

**Summary:**

Pipeline for an ASR-ready dataset using EU parliamentary speech in many languages. Generates fairly sizable data (61K hours) across multiple languages (22); 51K hours have character error rate <20%, yielding 19 languages with >1K hours @ <20% CER. Finetuning with the dataset shows significant accuracy gains (40% WER improvement) for open source models.

**Additional Feedback:**

I've identified what I consider to be a weakness in exposition/analysis (point #1). The authors claim "The balanced distribution across languages addresses the severe imbalance in current multilingual speech resources." This is an important accomplishment, but I'd like to see the data to support this conclusion. You can make that explicit comparison, so please do so.

For example, in table 4 you illustrate how many hours of aligned speech is created for each language, and how many hours have CER (0,10%), (10%,20%), (20%,30%), >30%. This is great. Could you please compare against SOTA? E.g., how many hours of Estonian speech exist already under permissive licenses? How many hours are <20% CER (authors identify this as a key threshold)?

I will change my rating if you can do this.

Also it looks like there is some license flagging by the automated tooling. Please fix that.

**Dataset Code Accessibility:**

Partly

**Dataset Code Comments:**

I'm not in a position to comment here.

**Ethical Considerations:**

No, there are no or only very minor ethics concerns

**Final Justification:**

The authors did a great job of addressing my concerns, and helped make it clear how much this advances SOTA.

**Limitations Weaknesses:**

1. The paper does a good job of showing the accuracy gains from finetuning, especially for low resource languages. It doesn't speak to the change in raw amount of hours of ASR-ready speech by language (relative to SOTA).

2. I’m unsure the impact this additional ASR data would have on pre-training a model. Is there some way we could illustrate that? Finetuning is not the same as pre-training in my mind.

3. Why does the pipeline perform poorly on Maltese (a lot of >30% CER)? Is it the source data or the pipeline? Help me understand.

4. The pipeline itself isn't particularly novel.

**Strengths Contributions:**

Very nicely explained paper, generally this is quite good work with good coverage of many languages and appears to significantly improve ASR performance for several under-served languages. Nice demonstration of impact via finetuning.

---

> ### Author Rebuttal · Authors · 2025-07-30
>
> We thank the reviewer for their thorough analysis and thoughtful feedback. Below, we address the points raised.
>
> > I’m unsure the impact this additional ASR data would have on pre-training a model. Is there some way we could illustrate that? Finetuning is not the same as pre-training in my mind.
>
> We decided to finetune existing models as the amount of data per language would not be sufficient to train good ASR models from scratch (for reference, whisper was trained on more than 680k hours of data). Finetuning is commonplace for ASR models, even when the pretraining dataset does not contain the target language, as it benefits model performance. In the end, we chose the setup that leads to the best models. Moreover, the finetuning results are sufficient to support our claim that our data quality is high enough to improve performance.
>
> > Why does the pipeline perform poorly on Maltese (a lot of >30% CER)? Is it the source data or the pipeline? Help me understand.
>
> The high CER is due to the model we use to generate our pseudo-labels (whisper-generated transcripts that we match with the human transcripts). We used a maltese finetune of whisper, which, while performing better than the baseline whisper, still significantly lags behind other languages. This means that the pseudo labels are often wrong and can not be successfully matched with the human transcript, which leads to more data being filtered out. That being said, the data that passes the filter is still of high quality, as demonstrated by our finetuning results.
>
> > The dataset consists entirely of parliamentary speech. This domain is characterized by formal, planned, and often repetitive language. Models trained exclusively on this data may not generalize well to more informal, spontaneous, and conversational use cases. Nowadays, in-the-wild data becomes more and more important.
>
> As we mentioned in the section on limitations, this is a valid concern. While not perfect, we evaluated our finetuned models on the widely-used Fleurs multi-lingual ASR dataset. The results show that at least to some degree fine-tuning on our data allows the models to generalize across different speech domains as the Fleurs dataset is not focused on parliamentary speech.
>
> > I've identified what I consider to be a weakness in exposition/analysis (point #1). The authors claim "The balanced distribution across languages addresses the severe imbalance in current multilingual speech resources." This is an important accomplishment, but I'd like to see the data to support this conclusion. You can make that explicit comparison, so please do so.
>
> > For example, in table 4 you illustrate how many hours of aligned speech is created for each language, and how many hours have CER (0,10%), (10%,20%), (20%,30%), >30%. This is great. Could you please compare against SOTA? E.g., how many hours of Estonian speech exist already under permissive licenses? How many hours are <20% CER (authors identify this as a key threshold)?
> I will change my rating if you can do this.
>
> We thank the reviewer for raising this point. We have included a table containing previous SOTA speech datasets using human transcripts for the languages we cover in the following:
>
>
> | Country/Language | SOTA Dataset Name | SOTA Dataset Hours | EuroSpeech Hours |
> |------------------|-------------------|-------------------|------------------|
> | Croatia | ParlaSpeech-HR [1] | 3061 | **5615.8** |
> | Denmark | FT-Speech [2] | 1800 | **5559.8** |
> | Norway | Stortinget Speech Corpus version 1.0 | 5190 | 3866.7 |
> | Portugal | Mosel [3] | 5492 | 3293.5 |
> | Italy | Mosel [3] | 3756 | 2813.7 |
> | Lithuania | Common Voice [4] | 25 | **2681.2** |
> | United Kingdom | Mosel [3] | 437238 | 2609.3 |
> | Slovakia | Mosel [3] | 61 | **2553.6** |
> | Greece | Yodas [5] | 126.75 | **2395.4** |
> | Sweden | RixVox-v2 | 22900 | 2312.8 |
> | France | Mosel [3] | 26984 | 2249.8 |
> | Bulgaria | BG-PARLAMA [6] | 249 | **2200.1** |
> | Germany | Mosel [3] | 9236 | 2184.4 |
> | Serbia | ParlaSpeech-RS [1] | 896.22 | **1855.7** |
> | Finland | Aalto Finnish Parliament ASR Corpus [7] | 3087 | 1848.2 |
> | Latvia | Common Voice [4] | 263 | **1218.8** |
> | Ukraine | Yodas[5] | 396.598 | **1191.1** |
> | Slovenia | ASR database ARTUR 1.0 | 884 | **1156.4** |
> | Estonia | TalTech Estonian Speech Dataset 1.0 | 1334 | 1014.9 |
> | Bosnia & Herzegovina | Yodas [5] | 9.36959 | **691.3** |
> | Iceland | Samrómur Milljón [8] | 967 | 647.4 |
> | Malta | MASRI | 44 | **613** |
>
> We created new SOTA duration datasets for 12 languages, crossing the 1k hour threshold for 8 languages where previous datasets were below this threshold. For 5 languages, our durations are 10 to 100 times greater than those of prior SOTA datasets. It is also important to highlight that some of the previous SOTA datasets are not as easily usable (i.e., they do not have a unified representation and cannot be used with a few lines of code from HuggingFace). Furthermore, the quality on some of these datasets is difficult to verify as they do not explain how they filtered the dataset. The EuroSpeech durations in this table refer to the 20% CER subset which is equivalent to the threshold used for VoxPopuli. We have added the information of this table to the revised manuscript.
>
>
> > Also it looks like there is some license flagging by the automated tooling. Please fix that.
>
> We thank the reviewer for pointing this out. We will update the license in the repository after the review period ends (NeurIPS rebuttal policy).
>
>
> Based on the provided feedback and suggested improvements to the manuscript, would the reviewer be willing to reconsider their score?
>
> ---
>
> [1]: Ljubešić, Nikola, Peter Rupnik, and Danijel Koržinek. "The parlaspeech collection of automatically generated speech and text datasets from parliamentary proceedings." International Conference on Speech and Computer. Cham: Springer Nature Switzerland, 2024.
>
> [2]: Kirkedal, Andreas Søeborg, Marija Stepanovic, and Barbara Plank. "FT Speech: Danish Parliament Speech Corpus." INTERSPEECH 2020. International Speech Communication Association (ISCA), 2020.
>
> [3]: Gaido, Marco, et al. "MOSEL: 950,000 Hours of Speech Data for Open-Source Speech Foundation Model Training on EU Languages." Proceedings of the 2024 Conference on Empirical Methods in Natural Language Processing. 2024.
>
> [4]: Ardila, Rosana, et al. "Common Voice: A Massively-Multilingual Speech Corpus." Proceedings of the Twelfth Language Resources and Evaluation Conference. 2020.
>
> [5]: Li, Xinjian, et al. "Yodas: Youtube-oriented dataset for audio and speech." 2023 IEEE Automatic Speech Recognition and Understanding Workshop (ASRU). IEEE, 2023.
>
> [6]: Geneva, Diana, Georgi Shopov, and Stoyan Mihov. "Building an ASR corpus based on Bulgarian Parliament speeches." International Conference on Statistical Language and Speech Processing. Cham: Springer International Publishing, 2019.
>
> [7]: Virkkunen, Anja, et al. "Finnish parliament ASR corpus: Analysis, benchmarks and statistics." Language Resources and Evaluation 57.4 (2023): 1645-1670.
>
> [8]: Mena, Carlos Daniel Hernández, Þorsteinn Daði Gunnarsson, and Jón Guðnason. "Samrómur milljón: An asr corpus of one million verified read prompts in icelandic." Proceedings of the 2024 Joint International Conference on Computational Linguistics, Language Resources and Evaluation (LREC-COLING 2024). 2024.

---

> > ### Comment · Reviewer_otqo · 2025-08-03
> > **Thanks for the excellent response**
> >
> > Thank you for your thorough and thoughtful response. I particularly appreciate the table you created showing improvement in hours of speech over SOTA. It would be fantastic if we could incorporate this table into the paper, e.g., as an appendix.
> >
> > I will increase my rating, congrats!

---

> > > ### Author Response · Authors · 2025-08-03
> > >
> > > We thank the reviewer for their constructive feedback and for increasing their score. It is greatly appreciated. We have incorporated the comparison table into the revised manuscript.

---

### Official Review · Reviewer_WyJC · 2025-07-02

**Rating:** 5
**Confidence:** 4

**Summary:**

This paper introduces EuroSpeech, a large-scale, multilingual speech corpus derived from the public recordings of 22 European parliaments. The primary contribution is a new dataset containing over 61,000 hours of aligned speech. The dataset provides substantial data for languages often considered low-resource, with 19 languages exceeding 1,000 hours and all 22 exceeding 500 hours. This paper also provides an open-source, scalable pipeline developed to create this dataset. The pipeline is designed to handle the complexities of real-world data, such as long-form audio and non-verbatim transcripts, featuring a robust two-stage alignment algorithm. This paper validates the dataset's utility by finetuning a pretrained ASR model (Whisper v3 Turbo) on six languages, achieving an average relative WER reduction of 41.8% on the FLEURS benchmark.

**Additional Feedback:**

I noticed that the paper includes only 24 references. In recent years, there has been a surge of ASR datasets related to speech-text alignment and Whisper-based pseudo-labeling. Some notable contributions in this area include Libriheavy [1] and GigaSpeech 2 [2]. Including a broader range of related work would strengthen the paper's context.

---
[1] W. Kang et al., "Libriheavy: A 50,000 Hours ASR Corpus with Punctuation Casing and Context," IEEE International Conference on Acoustics, Speech and Signal Processing (ICASSP), 2024, Seoul.

[2] Y. Yang et al., "GigaSpeech 2: An evolving, large-scale and multi-domain ASR corpus for low-resource languages with automated crawling, transcription and refinement," The 63rd Annual Meeting of the Association for Computational Linguistics (ACL), 2025, Vienna.

**Dataset Code Accessibility:**

Yes

**Dataset Code Comments:**

Dataset: https://huggingface.co/datasets/disco-eth/EuroSpeech

Code: https://github.com/SamuelPfisterer/EuroSpeech

**Ethical Comments:**

The authors have thoroughly considered and addressed the ethical dimensions of their work. They have a dedicated discussion in Section 6 and have completed the NeurIPS checklist with transparency.

**Ethical Considerations:**

No, there are no or only very minor ethics concerns

**Final Justification:**

The authors have addressed my concerns raised in the initial review. Specifically:

- They clarified the dependency of alignment quality on ASR model performance and explained the filtering mechanism that preserves high-quality data, even for low-resource languages.

- They acknowledged the limited number of references and have updated the manuscript to include additional relevant work, such as Libriheavy and GigaSpeech 2.

These clarifications and updates satisfactorily resolve the issues I raised. I have no remaining concerns, and my final score leans towards accept.

**Limitations Weaknesses:**

- As mentioned in Section 6, the alignment pipeline relies on an existing Whisper ASR model to generate pseudo-labels for matching. The quality of the final aligned dataset is therefore constrained by the performance of this initial model. For languages where the baseline ASR model performs poorly, the alignment accuracy and resulting data quality might be reduced.
- The dataset consists entirely of parliamentary speech. This domain is characterized by formal, planned, and often repetitive language. Models trained exclusively on this data may not generalize well to more informal, spontaneous, and conversational use cases. Nowadays, in-the-wild data becomes more and more important.

**Strengths Contributions:**

There's a lot to like in this paper, which stands to make it a valuable contribution to the speech processing community:
- The primary contribution is the EuroSpeech dataset, which provides over 61k hours of aligned speech. As shown in Table 1, EuroSpeech stands out not just for its total size but for its superior per-language coverage, providing over 1,000 hours for 19 languages.
- This dataset was downloaded 29,629 times last month. This seems to indicate that it is a popular dataset and its quality has received positive feedback from users.
- The paper introduces a well-engineered and open-source data processing pipeline. The design is modular, encompassing data sourcing, downloading, and alignment.
- The paper is well-written, clearly structured, and easy to follow.

---

> ### Author Rebuttal · Authors · 2025-07-30
>
> We thank the reviewer for their thorough analysis and thoughtful feedback. Below, we address the points raised.
>
> > As mentioned in Section 6, the alignment pipeline relies on an existing Whisper ASR model to generate pseudo-labels for matching. The quality of the final aligned dataset is therefore constrained by the performance of this initial model. For languages where the baseline ASR model performs poorly, the alignment accuracy and resulting data quality might be reduced.
>
> The observation that the pipeline relies on existing ASR models to generate pseudo-lables for matching is correct. The accuracy of the transcription (the pseudo-labels) depends on the ASR model used. However, as shown with low resource languages (e.g., Maltese), worse pseudo labels primarily lead to more data being filtered out because the pseudo labels can not be sufficiently matched with the human transcript. The quality of the data that passes this filter remains high. If one trains on the filtered data (e.g., <10% CER), like we did for Maltese, one can still see significant model improvements as shown in our paper.
>
> > I noticed that the paper includes only 24 references. In recent years, there has been a surge of ASR datasets related to speech-text alignment and Whisper-based pseudo-labeling. Some notable contributions in this area include Libriheavy [1] and GigaSpeech 2 [2]. Including a broader range of related work would strengthen the paper's context.
>
> We thank the reviewer for finding these missing references. We have updated the manuscript accordingly to include the mentioned references as well as others we found.

---

> > ### Comment · Reviewer_WyJC · 2025-08-03
> >
> > Thank you for the response and clarifications. I have no further concerns. My final score remains leaning towards accept. Good luck.

---

### Decision · Program_Chairs · 2025-09-18

**Decision:**

Accept (spotlight)

**Comment:**

The paper describes a pipeline to gather automatic speech recognition (ASR) data by mining parliamentary recordings and using Whisper to generate candidate transcriptions. While the use of parliamentary recordings is not novel, the authors demonstrate impressive results over the state of the art on multiple low-resource languages stemming from the collected data, providing at least three valid contributions to the track (the pipeline, the dataset, and the fine-tuning results).

After the rebuttal, all reviewers recommend acceptance (5, 5, 5). Two reviewers (WyJC, otqo) highlighted some concerns that were fully addressed in the rebuttal. These included a potential bias from collecting on a very specific type of data (which the authors address by providing transcription error rates on a different dataset), issues arising from using transcription models having potentially low accuracy on some languages (e.g., Maltese), and the lack of results when training from scratch.

One reviewer (YWXy) questioned the novelty of the dataset and the pipeline, while still providing a rating of 5. This reviewer did not engage in the rebuttal (despite several reminders). I personally find the answer from the authors convincing and, lacking any response from the reviewer, I will ignore these concerns.

All reviewers agree that the paper is interesting for the community and useful for low-resource languages. No reviewer has questioned the technical correctness of the results or the methodology. As I see no reason to go against the consensus of the reviewers, I recommend acceptance with no reservations.